# Preparation and Reaction Mechanism Characterization of Alkali-activated Coal Gangue–Slag Materials

**DOI:** 10.3390/ma12142250

**Published:** 2019-07-12

**Authors:** Hongqiang Ma, Hongguang Zhu, Cheng Yi, Jingchong Fan, Hongyu Chen, Xiaonan Xu, Tao Wang

**Affiliations:** School of Mechanics and Civil Engineering, China University of Mining and Technology (Beijing), Beijing 100083, China

**Keywords:** alkali-activated coal gangue–slag, paste fluidity, cement paste fluidity, compressive strength, polycondensation reaction

## Abstract

In this paper, slag is used as a calcium source to make alkali-activated coal gangue–slag (AACGS) based material. The reaction mechanism of AACGS materials was discussed in depth by means of XRD, FT-IR, ^29^Si MAS-NMR (nuclear magnetic resonance) and SEM-EDS (energy dispersive spectrometer). The experimental results show that coal gangue can be used as a raw material for preparing alkali-activated materials. The liquid–solid ratio is the most influential factor on AACGS paste fluidity and strength, followed by slag content. As the modulus of sodium hydroxide increases, the depolymerization process of the reactant precursor is accelerated, but the high sodium hydroxide concentration inhibits the occurrence of the early coal gangue–slag polycondensation reaction, and exerts little effect on the 28 d compressive strength. Ca^2+^ in the slag promotes exchange with Na^+^, and the product is converted from N-A-S-H gel to C-(A)-S-H gel, and C-(A)-S-H is formed with higher Ca/Si ratio with the increase of slag content. The slight replacement of coal gangue by slag can greatly improve the reaction process and the strength of AACGS materials.

## 1. Introduction

Cement is one of the world’s most widely used construction cementitious materials. About 1.6 billion tons of cement is used each year, releasing 1.5 billion tons of CO_2_, and about 5% to 8% of global CO_2_ emissions are caused by cement [1,2,3,4]. It has become crucial to develop new, sustainable and low-carbon construction materials with the advent of low-carbon economy. Among the numerous alternatives available, alkali-activated materials have aroused widespread concern, since they can be used as a 100% replacement to cement and significantly reduce the emission of CO_2_ and dust particles [5]. Alkali-activated materials are characterized by high initial strength, low hydration heat, freezing resistance, chemical resistance, strong durability and excellent high-temperature stability [6,7,8,9,10]. The alkali-activated materials can be considered as the best replacements for cement based on these advantages [11].

Alkali-activated materials are products of the reactions between alumino-silicate materials (fly ash, slag, metakaolin, coal gangue, ceramic, etc.) and alkaline solutions at ambient temperature [11,12]. All materials rich in active silicon-aluminum oxides (slag, fly ash, metakaolin, coal gangue, etc.) can undergo polymerization reactions with high-concentration alkalis [13]. However, the reaction products of alkali-activated slag (rich in CaO) are different from those of alkali-activated metakaolin, coal gangue or fly ash (rich in Si and Al) the former are C-(A)-S-H gels while the latter are N-A-S-H gels [14,15], the structure of which is influenced by alkali concentration and Si/Al molar ratio [16]. So far, studies on alkali-activated materials have been primarily focused on commonly used supplementary cementitious materials such as metakaolin, slag and fly ash.

Coal gangue is the solid waste produced in coal mining. In countries with developed coal resources, such as China, India, Australia, Germany, the United States and South Africa, the discharge of coal gangue waste remains high [17]. Waste storage generated from coal gangue has a high risk of environmental pollution [18]. The coal gangue crystalline mineral phase is resolved into SiO_2_ and Al_2_O_3_ components, which has intensive pozzolanic activities under high temperature [19,20]. Literatures on alkali-activated calcined coal gangue are deficient. Wan et al. [21] used CaO as the activator and calcined coal gangue, slag and gypsum to configure P.C 32.5 cementitious materials. Geng et al. [22] used red mud and coal gangue as raw materials to prepare binary geopolymers, the results show that the final product is composed of amorphous silica–aluminum base polymer gel and some impurity filler, which can be used as a new building material. Han et al. [23] prepared coal gangue–slag composite materials with sodium silicate as an activator, and found that the hydration products were mainly zeolite. The compressive strength of the composite system was greater than 40 MPa when the content of calcined coal gangue was less than 30%. Yi et al. [24] prepared coal gangue geopolymers with sodium hydroxide and sodium silicate as alkali activators, and found that coal gangue can be used as geopolymer raw material. However, compared with P.O 42.5 cement specimens, coal gangue geopolymers have higher 1 d compressive strength and lower 28 d compressive strength. However, Kumar et al. [25] significantly improved the low reactivity of alkali-activated fly ash by adding slag to fly ash. Therefore, the calcium source can be used as an important measure to improve the strength of alkali-activated materials.

In this paper, with silica–alumina coal gangue as the main material, slag as the calcium source, NaOH and Na_2_SiO_3_ to be the alkali activator, the AACGS materials is prepared. Evaluate the impacts of slag content (0%, 10%, 20%, 30%, 40% and 50%), NaOH molar concentration, NaOH/Na_2_SiO_3_ mass ratio and liquid–solid ratio on AACGS paste fluidity, non-evaporable water dosage, compressive strength and microstructure (XRD, FT-IR, ^29^Si MAS-NMR(nuclear magnetic resonance) and SEM-EDS (energy dispersive spectrometer)) through experiments. In-depth analysis of the reaction mechanism of AACGS materials.

## 2. Experimental

### 2.1. Raw Materials

The Shanxi clay coal gangue powder selected was calcined with constant temperature in muff furnace 700 °C for 2 h, the slag was the S95-grade granulated blast-furnace slag from Hebei Iron and Steel Company (Shijiazhuang, China). The 96% sodium hydroxide and a modulus of 3.22 (26.5% SiO_2_, 8.5% Na_2_O and 65% H_2_O) sodium silicate solution were used as the alkali activator.

The main chemical compositions of calcined coal gangue, S95-grade slag and cement are given in Table 1. It is obvious that calcined coal gangue belongs to the silica–alumina material, while slag belongs to the calcium–silica–alumina material. Figure 1 reveals the grain size distribution of coal gangue and slag, and the median particle size d50 values of coal gangue and slag were 17.252 μm and 10.529 μm, respectively.

### 2.2. Mixes Design and Specimens Preparation

In this paper, the effects of slag content, NH molar concentration, liquid–solid ratio and NaOH/Na_2_SiO_3_ mass ratio on AACGS materials were studied. The P.O 42.5 pure cement samples as a controlled sample, and the specific mixing ratio was displayed in Table 2, a total of 19 experiments were carried out. The slag content is 0%, 10%, 20%, 30%, 40% and 50% respectively; six kinds of liquid–solid ratio (liquid refers to the sum of all liquids in the mixed system, including the external addition of water, water in NS solution, and the H_2_O in NH solid. Solids refer to the sum of all solids in the mixed system, including coal gangue, fly ash, solids in NS solution and the mass of Na_2_O solids in NH solids) were selected, including 0.28, 0.30, 0.32, 0.34, 0.36 and 0.38; five different molar concentrations of NH were selected, respectively 8M, 10M, 12M, 14M and 16M; the mass ratios of NaOH/Na_2_SiO_3_ were 1:1, 1:1 1.5, 1:2 and 1:2.5 respectively. Sample numbers such as S0-12M2-36, where S0 represents 0% of the slag content, 12 represents the NaOH molar concentration, M2 represents the NaOH/Na_2_SiO_3_ mass ratio is 2 and 36 represents the liquid–solid ratio, which was 0.36.

The alkali activator solution should be configured 24 h before use, and NH solutions with different NH modulus should be configured respectively. After cooling to room temperature, the NS solution was mixed with the NS solution according to the mass ratio in Table 2. After mixing coal gangue and slag powder evenly, the alkali activator was added and stirred with cement mixer for 5 min, poured the fresh paste into steel molds quickly (40 mm × 40 mm × 40 mm) and vibrated them for 60 s on an electric vibration table to remove residual air. The molds were covered with thin polyethylene films and cured for 1 day at relative humidity (RH) = 95% ± 1% and T = 20 ± 2 °C. They were then demolded, transferred to standard curing rooms and cured for 3 d, 7 d and 28 days respectively.

### 2.3. Testing Methods

#### 2.3.1. Samples Performance Testing

(1) Paste fluidity: The paste fluidity of AACGS was measured by the standard net paste fluidity test model (upper diameter Φ = 60 mm; lower diameter Φ = 36 mm; H = 60 mm) [24].

(2) Compressive strength: According to GB/T 17671-1999 compressive strength test standard, test and calculate the compressive strength value. A YAW-300 pressure testing machine (Beijing, China) was employed to test 1 d, 3 d, 7 d and 28 d compressive strength of the cube specimens of AACGS materials. The loading speed ranged from 0.5 to 0.8 MPa/s. Three samples of each mixture proportion were tested, with the experimental values averaged to generate the test value for each mixture proportion. If the compressive strength value of one sample exceeded the median value by 15%, the median value was taken as the compressive strength value of the samples.

(3) Non-evaporative water: Non-evaporative water content was measured by combustion method [24]. A certain amount of samples were dried at 105 °C for 24 h and weighed. Then calcining at 900 °C for 3 h at constant temperature, the mass loss was used as the non-evaporable water content.

#### 2.3.2. Samples Microstructure Testing

The samples were cored, broken or crushed, terminated hydration with anhydrous ethanol and at 60 °C in a vacuum dried for 4 h. Samples with size of about 0.5–1 cm were selected for SEM-EDS test. The samples were grinded with agate mortar powder until they felt no granularness, and sifted by a 0.08 mm square-mesh sieve (particle size greater than 200 mesh). The following techniques were used to monitor the reactions, characterize the reaction products and analyze the microstructures:

(1) X-ray diffraction analysis: Using a Bruker D8 X-ray diffractometer (Bruker company, Germany) with CuKα target radiation (scanning speed = 2°/min; step size = 0.02°) to continuously scan specimens at 5° to 70° (2θ°).

(2) FT-IR analysis: Adopting a Nicolet 6700 FT-IR spectrometer (Nicolet company, USA, spectral range = 400–4000 cm^−1^; resolution = 0.09 cm^−1^; dynamic adjustment = 130,000 times/s) for the FT-IR test.

(3) ^29^Si MAS-NMR analysis: Bruker Avance III 600M solid-state high-resolution nuclear magnetic resonance spectrometer (Bruker company), recording NMR spectra of Si at 39.72 MHz.

(4) SEM-EDS analysis: The samples surfaces are sputter coated with gold-palladium prior to imaging. Employing a field emission scanning electron microscope (SU8010; Hitachi; Tokyo, Japan) to observe surface morphology of the broken specimens and using an energy spectrometer to quantitatively analyze microelement content of the specimens. For chemical compositions of the reaction products, SEM-EDX point analyses were used, and 30 points per sample was tested [26].

## 3. Results and Discussions

### 3.1. Paste Fluidity Analysis of the AACGS

The fluidity of AACGS paste is mainly manifested as viscosity. The stronger the paste fluidity is, the better the initial expansion and slump of concrete prepared under identical conditions shall be. The paste fluidity of different samples is shown in Table 2. Paste fluidity rose as the slag content increases. In contrast to pure coal gangue samples (S0-12M2-36), samples with the slag content of 10%, 20%, 30%, 40% and 50% enjoyed a 7.10%, 14.49%, 17.61%, 20.65% and 23.55% increase in fluidity respectively. Kramar et al. [27] found that alkali-activated slag mortar (180 mm) possessed a greater fluidity than fly ash mortar (152 mm) and metakaolin mortar (166 mm) under the same conditions, which was mainly attributed to the following reasons: Slag had no regular crystals, thereby having a larger surface area than coal gangue. In addition, slag required less water than coal gangue. Therefore, the replacement of coal gangue by slag increased the liquid–solid ratio. It was revealed in Table 2 that the liquid–solid ratio makes a great influence on the fluidity of AACGS paste, and the larger the liquid–solid ratio, the greater the paste fluidity. Neither NaOH molar concentration nor NaOH/Na_2_SiO_3_ mass ratio was influential on paste fluidity. The paste fluidity was about 160 mm. Furthermore, the paste fluidity of all alkali-activated samples was much greater than that of P.O 42.5 cement pastes.

### 3.2. Compressive Strength of the AACGS

Figure 2 reveals the compressive strength of the AACGS samples, the effects of slag content (Figure 2a), liquid–solid ratio (Figure 2b) and NaOH molar concentration (Figure 2c) were considered. As shown in Figure 2a, as the slag content increased, the compressive strength of AACGS samples increased at different ages. When the slag content was 50%, the compressive strength of AACGS samples increased rate the most (55.98%). The compressive strength of samples 1 d was about 50% of 3 d compressive strength, and the compressive strength of samples 3 d was about 85% of 28 d compressive strength, indicating the AACGS cementitious material samples could obtain higher early compressive strength. Figure 2a also shows the compressive strength of P.O 42.5 pure cement samples. It can be found that the appropriate quantity of slag could be used instead of coal gangue to make alkali-activated materials, and it could obtain a better compressive strength than that of P.O 42.5 Portland cement samples, especially the early compressive strength was obviously better than that of P.O 42.5 Portland cement samples.

Figure 2b shows the compressive strength of samples at different liquid–solid ratios. Compressive strength, which was significantly influenced by water dosage and liquid–solid ratio of the mixture, linearly reduced with the increment of liquid–solid ratio. High liquid–solid ratio increased the ratio of liquid to solid in the mixture and improved the workability and uniformity of alkali-activated materials. However, larger-volume activator solutions tend to cause large porosity [28]. Furthermore, it can be seen from Section 3.1 that the liquid–solid ratio was the most influential factor on paste fluidity. The massive fluid mediums in alkali-activated polymers prevent the contact between alkaline activators and source materials [29].

Figure 2c reveals the compressive strength of samples under different NaOH molar concentration. It can be found that the compressive strength of 1 d, 3 d, 7 d and 28 d was the highest when the NaOH molar concentration was 12 M, 14 M, 16 M and 16 M respectively. The high NH molar concentration will accelerate the polymerization reaction, leading to premature condensation of SiO_2_ and Al_2_O_3_, thus reducing the early compressive strength [30]. However, with the increase of curing age, the high-alkalinity environment inside the AACGS system still exists, and the polymerization reaction will continue, leading to the increase of the later strength. In comparison with samples whose NaOH = 8 M, samples with 10 M, 12 M, 14 M and 16 M enjoyed an increase in compressive strength by 3.55%, 4.30%, 5.51% and 7.07% respectively, which indicates that NaOH molar concentration made little difference on 28 d compressive strength when the slag content was 30%.

Figure 3 shows the compressive strength of samples at different NaOH/Na_2_SiO_3_ mass ratios. It was found out that the maximum compressive strength value was obtained under the circumstance that the NaOH/Na_2_SiO_3_ mass ratio valued 1:1.5. Both increased and decreased NaOH dosages decrease the compressive strength, reasons of which was below: Excessively low NaOH dosage generated the mixture insufficiently alkaline, thereby decreasing the polymerization rate and delaying the polymerization. If the NaOH dosage was excessively high, alkali-activated polymerization reacted rapidly, some coal gangue, slag and alkaline activators should fail to participate in the reaction and fill in the voids as residuals, thereby reducing the compressive strength.

### 3.3. Non-Evaporable Water Dosages of the AACGS

The non-evaporable water dosage is not only related to the amount of hydration products, but also concerned with the type of hydration products. The non-evaporable water dosage of pure cement samples is proportional to the amount of C-(A)-S-H gels. In this paper, hydration products of AACGS polymers were mainly structurally disordered and highly cross-linked three-dimensional N-A-S-H gels, C-A-S-H gels and alumina-silica network polymers. Figure 4a shows the non-evaporable water dosage of AACGS samples with different slag contents. It was found out that the non-evaporable water dosage linearly rose as the slag content increased. In contrast to pure coal gangue samples, with the slag content of 10%, 20%, 30%, 40% and 50% enjoyed a 7.74%, 15.02%, 21.48%, 31.26% and 34.94% increase in non-evaporable water dosage respectively, indicating that slag content had a great impact on non-evaporable water dosage. Figure 4b reveals the non-evaporable water dosage at different liquid–solid ratios. The non-evaporation water dosage first linearly increased with the liquid–solid ratio, and then slowly increased.

### 3.4. XRD Analysis of the AACGS

The main product of the AACGS samples is formed by intertwining the hydration products of the alkali-activated coal gangue and the alkali-activated slag. Figure 5 shows XRD patterns of AACGS samples with different slag contents, wider humps can be seen. The main characteristic peaks included SiO_2_, CaAl_5_Si_20_O_10_·4H_2_O, NaAlSiO_2_·H_2_O and AlSi_2_O_5_·xH_2_O, and the main crystalline phases were quartz and calcite. It can be seen from the figure that the cementitious materials contained a large amount of inert quartz that was not involved in chemical reactions. In other words, AACGS was a mixture of reactants and unreacted materials. Furthermore, the XRD diffraction pattern was analyzed by Jade, there were low-intensity diffraction peaks of calcium zeolite (Ca[Al_2_Si_3_O_10_]·3H_2_O) at 30° to 40° (2θ°; PDF #29-0809), reflecting the formation of a C-A-S-H gel phase. Moreover, the diffraction peak of SiO2 weakened and the diffraction peak of C-A-S-H gels were enhanced with increased slag content.

Figure 6 shows XRD patterns of samples under different NaOH/Na_2_SiO_3_ mass ratios. All the four samples had strong SiO_2_ and zeolite gel diffraction peaks and showed no difference in mineral composition. SiO_2_ with the NaOH/Na_2_SiO_3_ mass ratio of 1.5 had the lowest diffraction peak intensity. In other words, active SiO_2_ had higher reactivity, which is in accordance with the law of compressive strength. The crystalline phase of the zeolite appears in both Figure 5 and Figure 6, and the formation of the crystalline phase of the zeolite was positive for compressive strength of the samples. It has been found that the crystalline phase of the zeolite in the alkali-activated material is usually generated after mixing the raw material with the alkaline solution at high water dosage [31,32]. The study found that the amount of zeolite crystalline phase generated enhanced as the slag content increased.

### 3.5. FT-IR Analysis of the AACGS

Figure 7 shows the FT-IR spectra (spectral range = 400–4000 cm^−1^) of samples with the sample of S0-12M2-36 and S30-12M2-36. –OH bending vibration and H–O–H stretching vibration display absorption peaks at 3450 and 1650 cm^−1^ separately, which indicates that the samples contain chemically bound water [33]. Specifically, the absorption peaks appear at 3448.28 cm^−1^, 1641.77 cm^−1^, 3444.17 cm^−1^ and 1642.27 cm^−1^ in samples with the slag content of 0% and 30% respectively. 1429.86 cm^−1^ in Figure 7 is the absorption peak of the stretching vibration of C–O and no obvious C–O stretching vibration peak is detected in samples with the slag content of 0%. Studies have shown that the stretching vibration of C–O takes place at about 1440 cm^−1^ and relative absorption intensity of the peak increases with the increase in CaO dosage [34].

Figure 8 shows the FTIR spectra (spectral range = 400–4000 cm^−1^) of samples with different NaOH molar concentration. It was obvious to observe the most powerful vibration band that produces corresponding to the Si–O bonds of N-A-S-H and C-(A)-S-H gels, and wave number of this band depends on the precursor reactants [15,35]. In all samples, a strong peak at about 1010 cm^−1^ indicates the inhomogeneous stretching vibration of Si–O bond [14], indicating an important evidence of alkali-activated polymerization process [36]. Primary spectral bands of the FTIR spectra under 8 M, 10 M, 12 M, 14 M and 16 M were 1017.54 cm^−1^, 1015.93 cm^−1^, 1014.22 cm^−1^, 1009.22 cm^−1^ and 1008.73 cm^−1^ correspondingly. In other words, absorption peak of the Si–O bonds turned into low wave numbers with the increment of NaOH molar concentration, which suggests a lengthening of the Si–O bond, a reduction in the bond angle. The regularly arranged chain structure of Si–O bonds was broken and silicon was replaced by aluminum to form Si–O–Al bonds in alkaline atmosphere [37]. In Figure 7 where the slag content was 30%, the interactions between SiO_2_ and Al_2_O_3_ enhance with the increase in CaO dosage and absorption peak of the Si–O bonds turned into low wave numbers with increase in slag content. Along with the formation of new materials, the spectral band from 710 cm^−1^ to 720 cm^−1^ was formed by the bending vibration of Si–O–Al bonds caused by Al (IV) replacement of Si. Spectra band at about 570 cm^−1^ in the FTIR spectra was due to the stretching vibration of Si–O–Al bonds while that at 450 cm^−1^ was attributed to the in-plane bending vibration of Si–O bonds [38].

### 3.6. Si MAS-NMR Analysis of the AACGS

Figure 9 shows the ^29^Si MAS-NMR spectra for the AACGS samples. Chemical displacements are usually explained according to disparate silicon Q^n^ environments, where *n* represents the number of bridging oxygens of other Si atoms connected to each Si (SiO_4_) tetrahedral unit [39]. NMR analysis provides information on the location of the ^29^Si element, the coordination of the element and the degree of cross-linking of tetrahedrons to illustrate the polymerization process. The C-(A)-S-H gel mainly contains Q^1^, Q^2^(1Al) and Q^2^, which supports that Al can blend into the silicate chain without changing the structure in C-(A)-S-H. As the slag content increased, the dosage of active SiO_2_ decreased, resulting in the rise of Ca/Si ratio, and the chemical displacement of silica polyhedron moved towards the direction of low field, indicating that the degree of polymerization of silicate structure in the hydration products of AACGS system increased, and the silicate structure system with high degree of polymerization led to higher strength development, which is in accordance with the results of compressive strength.

### 3.7. SEM-EDS Analysis of the AACGS

Figure 10 demonstrates SEM-EDS analysis of the broken surface of AACGS paste. Three groups with slag content of 0%, 30% and 50% were selected for SEM-EDS characterization in order to better understand the influence of slag. As the figures indicate, the difference between the three samples microstructures was evident. Five main features were observed in three samples: (1) Micro-cracks, (2) N-A-S-H gels, (3) C-(A)-S-H gels, (4) agglomerates, and (5) unreacted coal gangue particles. Alkali-activated coal gangue samples (Figure 10a) had relatively poor surface conditions, with a small quantity of gelatinous and flocculent products. As the slag content increased, flocculent gel crystals on the section surface increased. The main polymerization product of the alkali-activated coal gangue sample is N-A-S-H gels and agglomerates (amorphous alumino–silicate gels); and the main polymerization product of AACGS is N-A-S-H gels, C-(A)-S-H gels and agglomerates. When coal gangue is partially replaced by slag, the polymer gels are more compact. The two pozzolanic materials are well combined under the action of alkali, thereby increasing the compressive strength of the mixture. Furthermore, uniform micro-cracks were observed in the samples, which were mainly due to the drying shrinkage of alkali-activated samples [40]. Melo et al. [41], Cartwright et al. [42] and Mastali et al. [43] showed that the shrinkage value of alkali-activated materials was larger than that of pure cement materials.

The chemical reaction of alkali-activated coal gangue–slag materials mainly involved five atoms, such as O, Si, Al, Ca and Na. Therefore, EDS only extracts the percentage of these five atoms (At%), and the average of the atomic percentage of each element in the three samples was calculated in Table 3. It is obvious that the Ca/Si ratio and (Ca + Na)/(Si + Al) ratio of AACGS samples increased significantly with the increase of slag content. The external active SiO_2_ and Al_2_O_3_ reacted with CaO to produce C-(A)-S-H gel with high Ca/Si ratio when Ca/Si was relatively high, which made the structure of slurry more compact and the macroscopic strength increased. Moreover, with the increase of slag content, the Si/Al ratio of AACGS samples increased, Si and Al elements in coal gangue and slag also controlled the whole alkali excitation reaction process. With the increase of slag content, Si/Al ratio increased, and higher Si/Al ratio would lead to higher strength growth [44]. In other words, the increment of slag content result in the generation of C-(A)-S-H gels with higher Ca/Si, Ca/Al and Si/Al ratios, thereby improving the density and compressive strength of samples.

## 4. Reaction Mechanism

Coal gangue belongs to silica–alumina material, which is decomposed by high temperature calcination to decompose SiO_2_ and Al_2_O_3_ with volcanic ash activity. The slag is a high calcium auxiliary cementing material with strong volcanic ash activity. Different from cement samples, AACGS materials were influenced by slag content, liquid–solid ratio, NaOH molar concentration and NaOH/Na_2_SiO_3_ ratio in respect to the mechanic structure and microstructure significantly. Amorphous or metastable phases in silicon–aluminum materials chemically react with alkaline activators to form dense composites materials while crystalline phases fill internal voids of the materials [45]. Alkali-activated silica-alumina based materials are amorphous three-dimensional network structures [Q4(Al)] composed of SiO4 and AlO4 tetrahedrons that are linked by shared oxygen atoms [28].

In this paper, Al_2_O_3_ dosage of the coal gangue was up to 36.78%. Al_2_O_3_, which was obtained through the dehydroxylation decomposition of kaolinite, and had enormous pozzolanic activity, supposed to resemble the properties of metakaolin [46]. All Al_2_O_3_ in slag participated in geopolymerization, as the slag content increased, the Al_2_O_3_ dosage in the mixture reaction precursor decreased. In the reaction process of AACGS materials, Ca^2+^ in the slag played an important role in the strength development. –OH attacked vitreous bodies of coal gangue and slag, leading to the fracture of vitreous Al-O-Si bond and the depolymerization and separation of the silica–aluminum phase. Furthermore, with the decrease of Al_2_O_3_ dosage, the alkali-activated geological polymerization process was divided into highly polymerized AlO4 and SiO4 units and a quasi-duplex structure rich in highly depolymerized SiO4 units. The atomic potential energy of Si^4+^ was twice in the tetracoordinate that of Al^3+^, exhibited stronger acidity and more easily polarized electrons on adjacent alkali metal oxide ions [47]. Na^+^ and Ca^2+^ are used to balance the charge, Ca^2+^ has a stronger driving adsorption ability than Na^+^, and the high charge density of Ca^2+^ drives the precipitation of the crystal phase of the C-(A)-S-H gel [35], resulting in an increase in compressive strength.

Bernal et al. [48] found that low MgO dosage facilitated the integration of Al–O tetrahedrons into the C–S–H chain. Therefore, the slight replacement of coal gangue by slag not only introduces CaO, but also changes the dosage of Al_2_O_3_ and MgO in the mixture. The increase in slag content increased the dosage of CaO and MgO in the mixture, decreased the dosage of Al_2_O_3_, enhanced the reactivity of alkali reactions, decreased the chain length of (N,C)-(A)-S-H phases and increased the Si/Al ratio, which was perfectly in compliance with the SEM-EDS analysis.

The results of XRD analysis showed that an amorphous silicon–aluminum network similar to the zeolite crystal phase was formed in the AACGS sample. The product of the S0-12M2-36 sample was zeolite-like alkali metal aluminosilicate (mainly sodium alumino-silicate). After incorporation of the slag, a calcium alumino–silicate product (C-(A)-S-H gel) appeared in the sample. The formation of three-dimensional amorphous polymer cementitious materials conduced to the binding performance of alkali-activated materials. The activation process of alkali-activated coal gangue is as follows: Active SiO_2_ and Al_2_O_3_ in coal gangue dissolve under the action of strong alkali, followed by the polycondensation of silicon aluminum. Sodium ion that is used to balance the charge is absorbed into silica–alumina gel phases.

In the alkali-activated coal gangue–slag reaction system, the coal gangue is the silica–aluminum material and the slag is the calcium–silica–aluminum material. CaO content in the alkali-activated coal gangue–slag reaction system increased with the increase of slag content. CaO reacts with different monomers (include silicate monomer and aluminate monomer) to form different reaction products. Some CaO reacts with silicate and aluminate to form the C-A-S-H gel, and some CaO reacts with silicate monomer to form the C–S–H gel. Some CaO reacted with the dissolved silicate and aluminate, which formed C–S–H gels. Some Ca^2+^ displaced Na^+^ to form C-A-S-H gels. The reaction products of alkali-activated coal gangue-slag materials were C-(A)-S-H gels, N-A-S-H gels and silico-aluminate colloid, and the three gels interweave to form the network disordered structure of alkali-activated coal gangue-slag, which had good compatibility [2,49].

## 5. Conclusions

This paper explored the compressive strength and reaction mechanism of AACGS polymers and discovered that the strength and hydration were influenced by multiple factors, including the slag content, NaOH molar concentration, NaOH/Na_2_SiO_3_ mass ratio and liquid–solid ratio. Specific conclusions drawn from this research are as follows:

(1) Liquid–solid ratio is the most influential factor on the fluidity and compressive strength of AACGS paste, followed by slag content. With the increase of liquid–solid ratio, the net paste fluidity of AACGS samples increased, but the compressive strength value decreased significantly. With the increase of slag content, the net paste fluidity and compressive strength of AACGS samples increased.

(2) NaOH molar concentration makes little difference to paste fluidity and 28 d compressive strength, high-molar concentration NaOH inhibits early polymerization of AACGS paste. The NaOH/Na_2_SiO_3_ mass ratio also has little influence on paste fluidity of the samples.

(3) Main hydration products of AACGS materials include N-A-S-H gels, C-A-S-H gels and some other alkaline aluminosilicate gels. As the slag content increases, C-(A)-S-H gels with high Ca/Si, Ca/Al and Si/Al ratios are generated, thereby increasing the compactness of the AACGS materials.

(4) The finding of this study demonstrated that adding a small amount of slag could improve the compressive strength of alkali-activated gangue obviously. The mechanical properties of AACGS have been obtained under different conditions, and the polymerization mechanism of AACGS has been studied in depth. The application of coal gangue materials was expanded in this paper, which provided factual basis for the preparation of polymer materials by using coal gangue. This study expanded the application of coal gangue in cementing materials, but its long-term drying shrinkage and durability need to be further studied.

## Figures and Tables

**Figure 1 materials-12-02250-f001:**
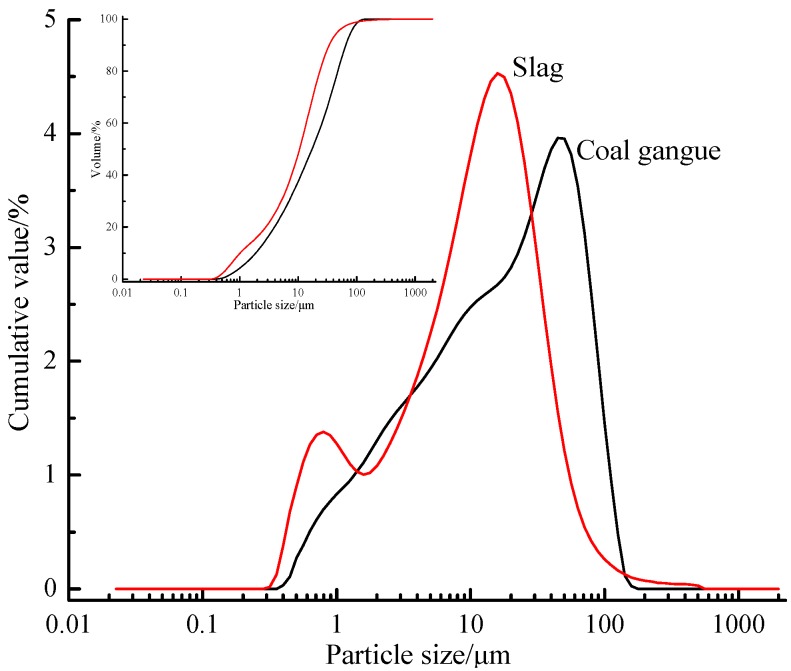
Calcined coal gangue and slag particle size analysis.

**Figure 2 materials-12-02250-f002:**
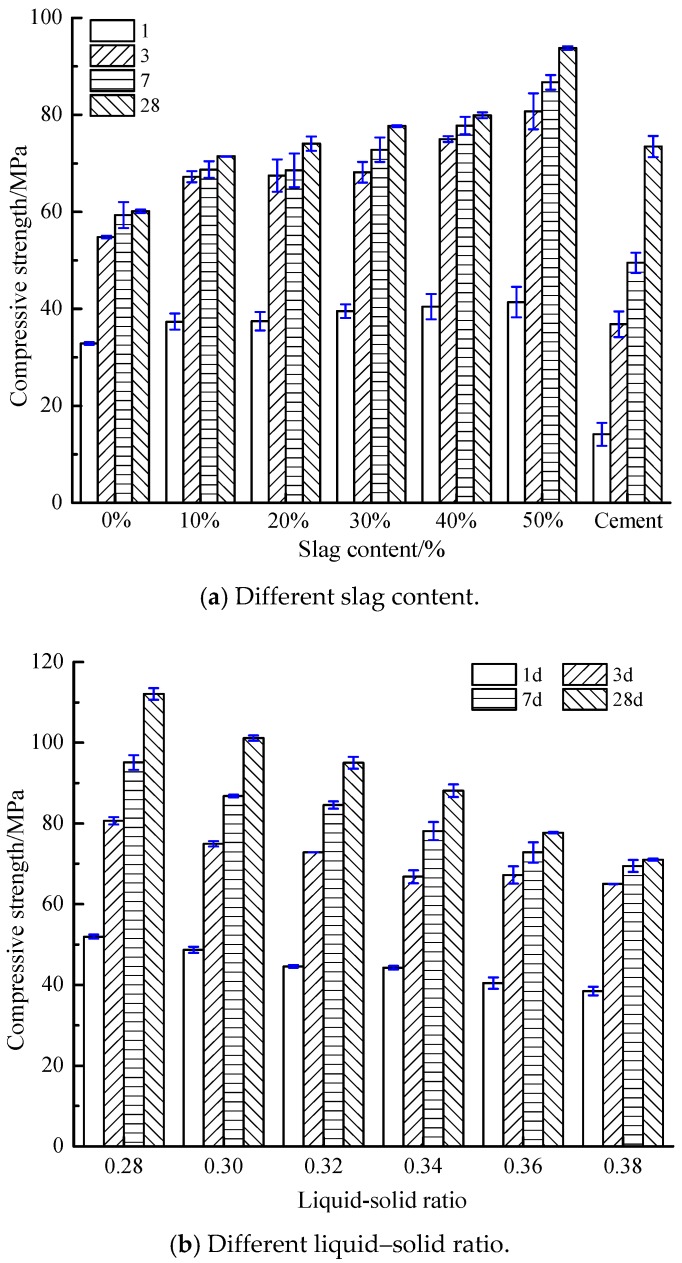
Compressive strength of AACGS samples. (**a**) Different slag content; (**b**) Different liquid–solid ratio; (**c**) Different NaOH molar concentration.

**Figure 3 materials-12-02250-f003:**
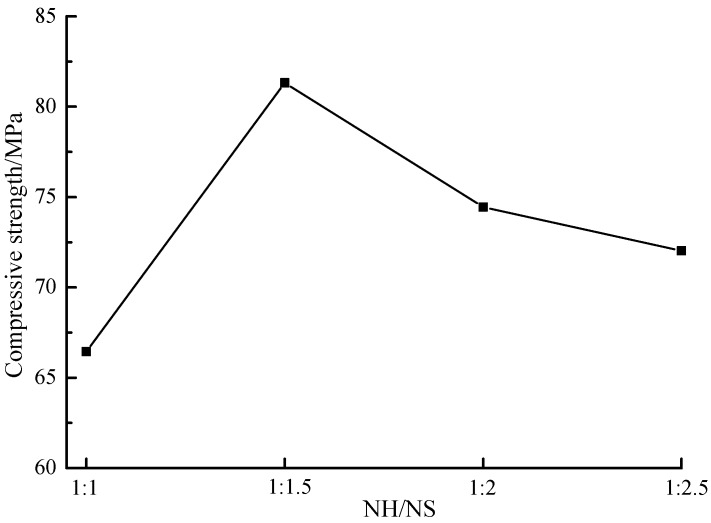
The compressive strength of samples with different NaOH/Na_2_SiO_3_ mass ratio. (NaOH = 8 M).

**Figure 4 materials-12-02250-f004:**
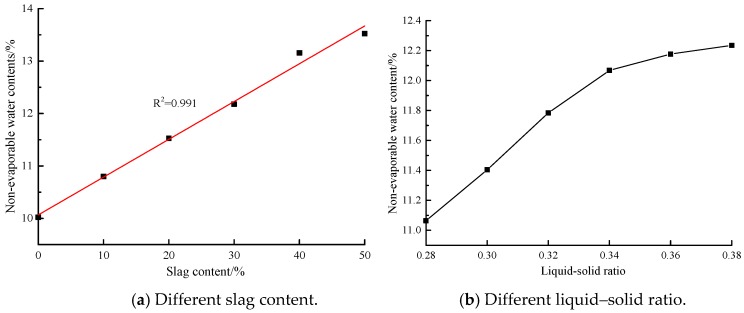
The non-evaporable water dosages of AACGS samples. (**a**) Different slag content; (**b**) Different liquid–solid ratio.

**Figure 5 materials-12-02250-f005:**
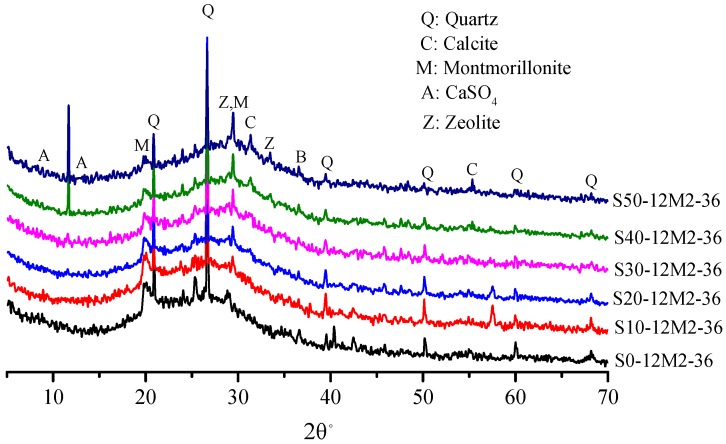
The XRD patterns of AACGS samples with different slag content.

**Figure 6 materials-12-02250-f006:**
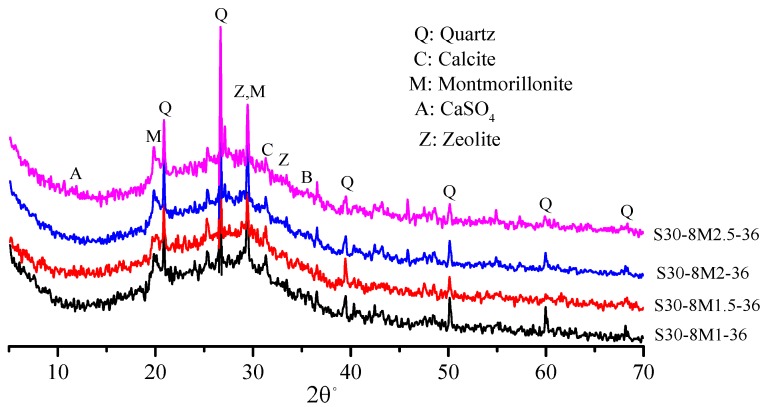
The XRD patterns of AACGS samples with different NaOH/Na_2_SiO_3_ mass ratio.

**Figure 7 materials-12-02250-f007:**
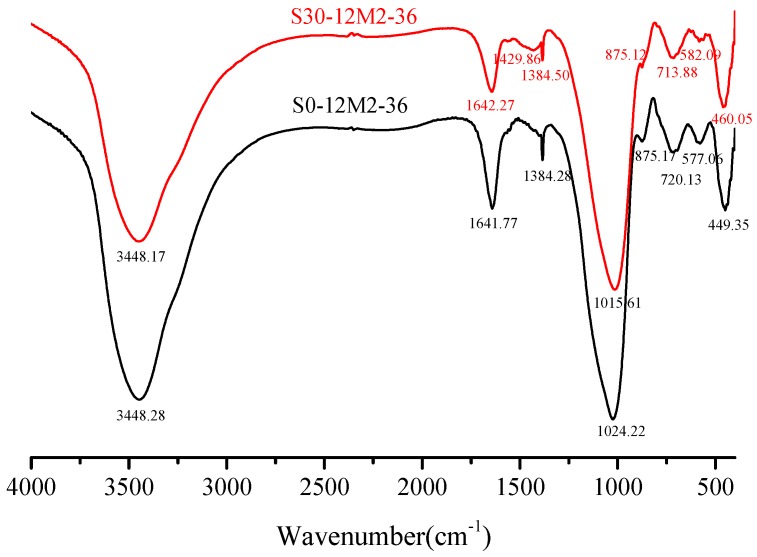
FTIR characterization of hardened paste with different slag contents.

**Figure 8 materials-12-02250-f008:**
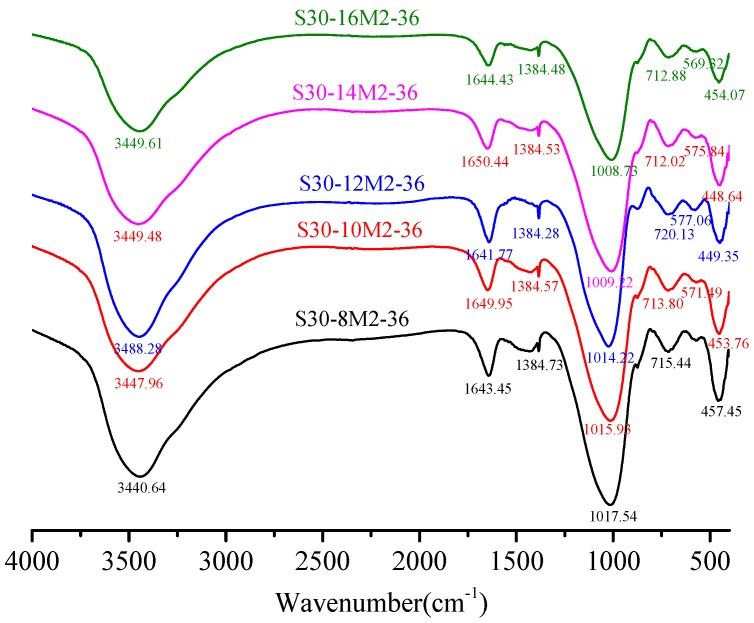
FTIR characterization of hardened paste with different NaOH molar concentration.

**Figure 9 materials-12-02250-f009:**
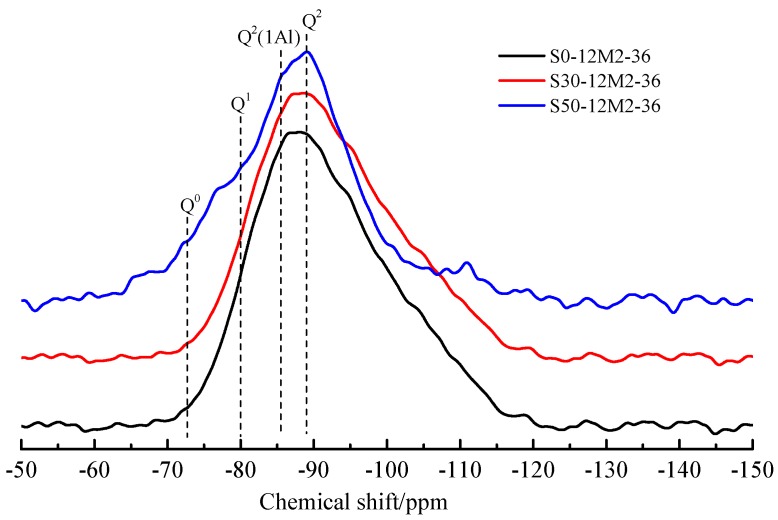
^29^Si MAS-NMR spectra for AACGS samples.

**Figure 10 materials-12-02250-f010:**
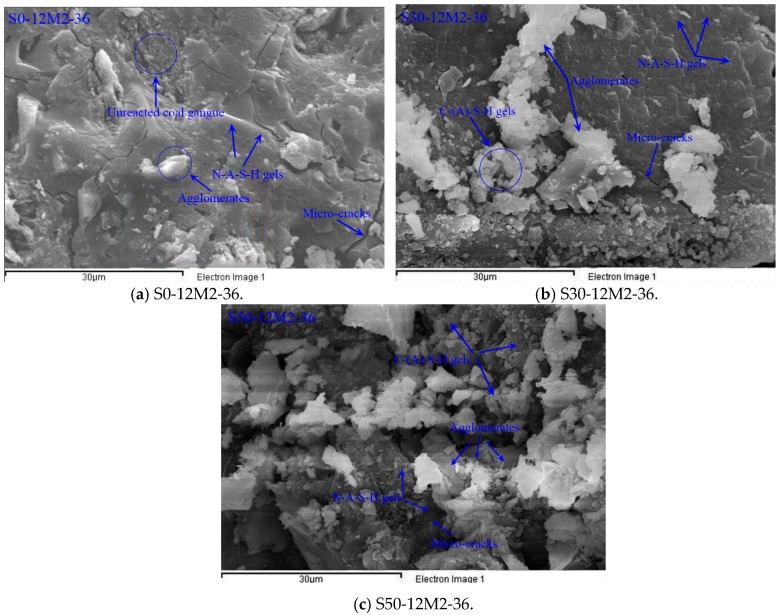
SEM-EDS analysis of the broken surface of the AACGS paste. (**a**) S0-12M2-36; (**b**) S30-12M2-36; (**c**) S50-12M2-36.

**Table 1 materials-12-02250-t001:** Chemical component of calcined coal gangue, slag and cement (mass fraction, %).

Parameters	Calcined Coal Gangue	Slag	Cement
SiO_2_	56.56	30.58	20.88
Al_2_O_3_	36.78	14.04	5.57
CaO	0.62	38.43	62.09
Fe_2_O_3_	1.95	0.35	2.40
MgO	0.22	10.57	2.43
Na_2_O	0.42	0.57	0.32
SO_3_	0.03	2.36	5.02
TiO_2_	2.10	1.93	0.31
LOI	1.32	1.17	0.98

**Table 2 materials-12-02250-t002:** Mix proportion of the alkali-activated coal gangue–slag (AACGS) paste.

Samples	Liquid-Solid Ratio	NaOH Molar Concentration (M)	NaOH/ Na_2_SiO_3_ ^a^	Binder Coal Gangue: Slag ^a^	Paste Fluidity/mm
Cement	0.36	—	—	—	60.0
S0-12M2-36	0.36	12	1:2	100:0	138.0
S10-12M2-36	0.36	12	1:2	90:10	147.8
S20-12M2-36	0.36	12	1:2	80:20	150.8
S30-12M2-36	0.36	12	1:2	70:30	162.3
S40-12M2-36	0.36	12	1:2	60:40	166.5
S50-12M2-36	0.36	12	1:2	50:50	170.5
S30-12M2-28	0.28	12	1:2	70:30	110.0
S30-12M2-30	0.30	12	1:2	70:30	122.0
S30-12M2-32	0.32	12	1:2	70:30	136.0
S30-12M2-34	0.34	12	1:2	70:30	146.0
S30-12M2-38	0.38	12	1:2	70:30	172.0
S30-8M2-36	0.36	8	1:2	70:30	160.8
S30-10M2-36	0.36	10	1:2	70:30	160.5
S30-14M2-36	0.36	14	1:2	70:30	157.3
S30-16M2-36	0.36	16	1:2	70:30	160.5
S30-8M1-36	0.36	8	1:1	70:30	155.0
S30-8M1.5-36	0.36	8	1:1.5	70:30	152.0
S30-8M2.5-36	0.36	8	1:2.5	70:30	163.0

^a^ mass ratio.

**Table 3 materials-12-02250-t003:** The average of the atomic percentage of each element in the three samples (At%).

At%	Si	Al	Ca	Na	O	Ca/Si	Si/Al	(Ca + Na)/(Si + Al)
S0-12M2-36	24.64	18.45	1.65	5.14	50.12	0.067	1.336	0.158
S30-12M2-36	24.02	16.14	6.97	2.18	50.69	0.290	1.488	0.228
S50-12M2-36	22.02	15.03	10.76	1.35	50.84	0.489	1.465	0.327

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
