# Peer review of "Preparation and Reaction Mechanism Characterization of Alkali-activated Coal Gangue–Slag Materials"

_materials, 2019, doi:10.3390/ma12142250_

Round 1
Reviewer 1 Report
Title: Study on the Preparation and Performance of Alkali-activated Coal Gangue-Slag Cementitious Materials
The paper investigates the influence of slag substitution and l/s ratio on performance of alkali-activated Coal Gangue-Slag paste. A topic is of interest for the readers of Materials. However, interpretation of results is limited and there are many questions with regard to test methods and results which suggest major revision of this paper.
The main impressions are:
References’ list is very poor. More references are needed to be included, because there is a plenty of studies which studied similar behavior of alkali activated slag pastes such as in this paper. I suggest to authors to find them and add them to their work and reinforce better their introduction.
missing information about sample preparation in Materials and Methods section. Please, add them.
in-depth analysis and discussions of the results. Bonding mechanism is not enough studied. And I am not sure if that was the aim.
not sufficient experimental details and discussions. In all your Results section you do not compare your results with results obtained from literature. Please, compare your results with other researches. What did you find different, compared to available literature? Why your results are important? What they show compared to other people’s papers?
Abstract:
Please add the element for which MAS-NMR testing was done.
Introduction:
Introduction has to be enriched with other references related to the performed research of authors.
Materials and methods:
What is the Sulfur content of slags? Please, report the contents of S.
Why LOI is so high for slag (3.53)?
Please, add particle size distribution and chemical composition of your Cement P.O 42.5.
Table 4 What is macrography tests? What is microcosmic tests? What is cosmic? Why macrography?
The curing of the samples is the same as it is in [25]. Please, write it also in your paper. Readers are not going to spend time looking in other papers to find this information. Why did you choose the same curing as in ref [25]? Please, write your explanation in the paper.
What are the standard curing rooms in paper [25]? What conditions are in these rooms? Temperature? Relative humidity? CO2? Please, state all these.
Please write: alkaline activators, and not alkali activators.
What is liquid? Is that only water? Is that alkali + water? Please, make clear that readers know what is exactly L and what is S?
Please, explain how did you mix and make pastes, which sample size was cast for XRD, MAS-NMR, FTIR?
Which standard was used to test compressive strength of the pastes?
Do you have a photo of the set up for mini-flow test and paste in fresh state? It would be good to add it to the paper to illustrate how did you perform fluidity test? Why do you call it actually “fluidity test”? This is just a workability test what you performed in your work.
What was the setting time of different pastes? Why you didn’t measure this? Why you didn’t look at heat of hydration of your pastes? It would tell you a lot about the kinetics of the reactions in your systems.
What is bonding mechanism? Why did you call it as such? What is bonding in the paste? It is rather formation of reaction products that you have studied. Make this clear.
Please, can you explain how did you prepare you samples for SEM-EDS analysis?
You can clearly see at the surface of your sample carbonates (Fig 10) which are due to carbonation of not well dry sample (you can see formation of calcite from your XRD results). You have to repeat SEM-EDS analysis. Otherwise, these quality of analysis and photos is not acceptable.
Please, can you explain how did you prepare you samples for XRD analysis? How fine (what was the mean particle size of paste powder) were the samples? How did you grind and mill the pastes for XRD powder analysis?
Please, can you explain how did you prepare you samples for MAS-NMR analysis?
Please, can you explain how did you prepare you samples for FTIR analysis?
Fig 3 please add standard deviations on the plots for compressive strength results.
Fig 9 why did you choose these samples for MAS-NMR study?
Fig 10 Why did you make spectra as you did in your photos? Why not point analysis? Why not spectral imaging? Based on one spectrum, you cannot say nothing about composition of the reaction products in your samples. You need deeper analysis per sample. In which mode photos were obtained? BSE? SE? Why cracks are there in the samples? You need to perform better SEM-EDX analysis, current quality is extremely low and not satisfactory for scientific explanations.
Conclusions
Hydration of AACGS polymers? Please make distinction between cement that can hydrate and alkali activated materials which are not only hydrating but polymerizing and condensing.
What is an appropriate amount of slag instead of coal gangue that provides additional CaO?
Main hydration products of AACGS polymers include N-A-S-H gels, C-A-S-H gels and some other alkaline aluminosilicate gels.
What are other alkaline aluminosilicate gels?
How did you identify N-A-S-H gels? Where are detailed EDX analysis?
How did you study the charge density of the cation?
Author Response
Thanks to Reviewer for their affirmation and suggestion of the manuscript. Moreover, we are very sorry for the difficulties our writing has brought to the reviewers. We revised it according to your suggestions and replied to your questions. In the introduction section, we highlight the research background and significance, and supplement the references related to the research content of this paper. We have supplemented the experimental process and curing conditions, and we have also proofread and polished the conclusion, making the study of this manuscript more meaningful. Other modifications details are marked in red in the manuscript. Thank you for your time and recognition. I look forward to the opportunity to discuss academic topics with you. thank you!

Reviewer 2 Report
The authors present a paper on the alkaline activation of Coal Gangue-Slag. The work presented is well organized and well structured, however, it is not a very innovative work in this area of knowledge. There are already countless works on the subject with published and validated results. It is essential that the authors present in the initial part of the paper a plausible justification for the accomplishment of this work, as well as their contribution to the current state of knowledge on the subject.
Abstract and Introduction
In general, both the Abstract and the Introduction are well organized and contain the necessary and sufficient information for this type of work. However, some essential information needs to be presented, namely:
· Place the question addressed in a broad context and highlight the purpose of the study;
· Highlight the importance of the study and briefly place the study in a broad context;
· Present the contribution of this work to the current state of knowledge on the subject;
· Considering the differences in SiO2 and Al2O3 content % between the gang coal and the slag what are the expectations in terms of final results by replacing each other?
Experimental
In general, the authors summarize the materials and the methods used. There are, however, some doubts on the subject, namely:
· A brief description of the manufacturing and mixing process of the samples studied could be presented, only the bibliographic reference seems to me little given the importance of the process.
· What was the time and temperature of the sample curing process?
· What is the reason for using mostly a NaOH molar concentration of 12?
· What is the justification for the remaining NaOH solution molarity (M) concentrations, namely 8, 10, 14 and 16?
· Would not it be equally important to evaluate the shrinkage of the samples studied?
Results and discussions
In general the results are presented in a correct and perceptible way and the observed trends are explained and whenever possible are compared with the work of other authors. However, some doubts arise, namely:
· In line 101, authors refer to table 2, but should probably be table 3!
· Would not it be possible to cross-reference the results obtained with XRD and FT-IR?
· The analysis of SEM-EDS results is somewhat reductive! Only have the 3 images displayed?
Author Response

(The authors gave the same response as above.)

Reviewer 3 Report
1. A proofreading needs to be done by a native.
2. Check references 5 and 17! to be sure that coal gangue has been mentioned in the paper.
3. The authors could cite following papers:
https://www.sciencedirect.com/science/article/pii/S0921344917303361
https://www.sciencedirect.com/science/article/pii/S0950061818323043
https://link.springer.com/article/10.1007/s12649-019-00626-9
4. in the introduction the authors mentioned that solid activator can works better than liquid why they did not use powder in their mixes? Use this reference
(https://www.sciencedirect.com/science/article/pii/S0008884617306877)
5. Presented resulted in figures 2a, if gouge is the main binder, mixture 0% should be changed to 100% and the content of GGBFS slag should reduce this amount. For instance, the first columns are related to 100% gauge instead of 0%, the second series will be changed to 90%, and this reduces up to 50%.
6. Please explain why you have selected the mix with 30% gouge slag while even 50% has high strength (Figure 2b and 2c)?
7. In figure 4, how non-evaporated water was measured?
Author Response

(The authors gave the same response as above.)

Round 2
Reviewer 1 Report
Review
Dear Authors,
Thank you very much for your work on the significant improvement of your manuscript. However, still there are few issues which should be addresses before this manuscript is accepted for publication. The comments are given below.
Please, remove cementitious materials from the title and the whole manuscript. Alkali activated materials are not “cementitious materials”.
Please, correct title to be more specific toward the aim of your work.
Title: Study on the Preparation and Performance of Alkali-activated Coal Gangue-Slag Cementitious Materials
Such as:
“Performance of Alkali-activated Coal Gangue-Slag Materials” or
“Use of Coal Gangue-Slag as precursor for alkali activated materials” or something similar.
Abstract:
This sentence below is a conclusion from your work and you should remove this sentence to the end of the abstract:
“Coal gangue can be used as raw material for preparing alkali-activated cementitious materials.”
And your abstract can begin as:
In this paper, slag is used as calcium source to make alkali-activated coal gangue-slag(AACGS) based binary material. The polymerization reaction mechanism of AACGS cementitious materials was discussed in depth by means of XRD, FT-IR, MAS-NMR and SEM-EDS. The experimental results show that liquid-solid ratio is the most influential factor on cement AACGS paste fluidity and strength, followed by coal gangue-slag slag dosage. As the modulus of sodium hydroxide increases, the depolymerization process of the reactant precursor is accelerated, but the high sodium hydroxide concentration inhibits the occurrence of the early coal gangue-slag polycondensation reaction, and exerts little effect on the 28 d compressive strength. Ca2+ in the slag promotes exchange with Na+, and the product is converted from N-A-S-H gel to C-(A)-S-H gel, and C-(A)-S-H is formed with higher Ca/Si ratio with the increase of slag dosage. The slight replacement of coal gangue by slag can greatly improve the hydration reaction and the strength of AACGS cementitious materials.
Since you are dealing with highly sensitive analyses (XRD, FT-IR, MAS-NMR and SEM-EDS) of the material, you have to provide adequate method for reaction stoppage. Based on your description, you have used anhydrous ethanol and dried at 105℃ under nitrogen environment for 24h. This is not acceptable. I would like to draw your attention to possible loss of hydrated water from the reaction products (gels) and possible polymerization of AACGS (due to high temperature used, 105℃). This can lead to misleading results and conclusions. More about sample preparation, authors are kindly asked to read this reference below, where isopropanol is highly recommended to be used to stop the reaction stoppage. Ref: K. Scrivener, R. Snellings, B. Lothenbach A practical Guide to Microstructural Analysis of Cementitious Materials CRC Press (2016)
There are two options:
1. You repeat your all experiments with correct sample preparation using isopropanol (XRD, FT-IR, MAS-NMR and SEM-EDS)
2. Or you discussed why did you use your way of reaction stoppage and mention possible consequences of your sample preparation, so that readers are aware of this when they are looking and reading your results.
You have to be honest and if you choose for second option, you have to carefully address abovementioned issues.
Please provide two SEM photos in BSE mode, one of unreacted coal gangue slag and of unreacted blast furnace slag, so that we can see the shape and size difference of both materials. This will contribute to understanding of your raw materials (precursors).
Line 119 Which standard was followed for compressive strength tests? Please, add to section 2.3.1.
Title of the Table 3 is not complete.
Table 3 The specific microstructure testing methods for sample ?
It should be probably written
Table 3 The material chemical properties and microstructure tests
Microstructure is not tested with XRD, FT-IR, NMR, but reaction products of your systems were studied for chemical characterization. Microstructure is studied here with SEM tests.
Line 132 Samples with particle size of about 0.5-1 cm were selected for SEM-EDS test.
It’s not particle size, but sample size, please, correct as follows:
Samples with size of about 0.5-1 cm were selected for SEM-EDS test.
Particle size cannot be about 0.5-1 cm. Particle size can be below 100 µm (according to Figure 1).
Line 302 3.7 SEM-EDS analysis of the AACGS
This analysis and photos are not acceptable. Please read reference below for preparation of the samples for SEM-EDS analysis and provide new polished surfaces obtained in Backscattered electrons (BSE) mode. For EDX analysis you need multiple points for element analysis, and also for this issue, please, carefully read the reference, where you can see the points collection and way of data analysis.
Nedeljković, M., Šavija, B., Zuo, Y., Luković, M., & Ye, G. (2018). Effect of natural carbonation on the pore structure and elastic modulus of the alkali-activated fly ash and slag pastes. Construction and Building Materials, 161, 687-704.
Please provide also a SEM photo where you will indicate with arrows each of these components:
1. Unreacted particles (coal gangue slag, slag)
2. Reaction products (C(A)SH gel)
3. Pores
4. Microcracks
Readers should see a difference between coal gangue slag and blast furnace slag in your work, and how gel is distributed around this particles.
Line 343 Instead of cementitious mechanism, I would suggest using “reaction mechanism”.
“Cementitious mechanism” can be used ONLY for cement-based materials.
here, you have alkali activated materials, thus, it is better to use term “reaction mechanism”.
Line 377 SEM-EDX analysis can not show an amorphous silicon aluminum network similar to the zeolite crystal phase was formed in the AACGS sample.
Only XRD can show this result.
Line 387 What do you mean with this sentence “With the increase of slag content, a great deal of CaO is led to the mixture system”?
Line 388 Is it C-S-H gel or C-(A)-S-H gel? Please be consistent with type of the gel which is formed in the samples.
Line 393 Please refine this sentence. You didn’t study hydration but reaction mechanism in AACGS samples. Is this correct?
Author Response
Thank you very much for your suggestions and help from the reviewer. Thank you for your recognition of my revised manuscript. According to the suggestions of reviewer, we re-tested the SEM-EDS of the three samples, and modified the abstract section, the experimental section and the SEM-EDS analysis section carefully. Thank you for your patient guidance. We are looking forward to your confirmation. We are looking forward to it. Revised portion are marked in red in the manuscript. Once again, thank you very much for your comments, suggestions and help. We hope that the revision is acceptable and look forward to hearing from you soon.

Reviewer 2 Report
The authors responded adequately to the questions posed by the reviewer and amended the paper as requested. In these circumstances, I am of the opinion that the article should be considered for publication.
Author Response
Thank you very much for your recognition of the manuscript. I look forward to the opportunity to have academic exchange and cooperation with you.